# How to model Human Actions distribution with Event Sequence Data

## Abstract

This paper studies forecasting of the future distribution of events in human action sequences, a task essential in domains like retail, finance, healthcare, and recommendation systems where the precise temporal order is often less critical than the set of outcomes. We challenge the dominant autoregressive paradigm and investigate whether explicitly modeling the future distribution or order-invariant multi-token approaches outperform order-preserving methods. We analyze local order invariance and introduce a distribution-based metric to quantify temporal drift. We find that a simple explicit distribution forecasting objective consistently surpasses complex implicit baselines. We further analyze the emergence of *mode collapse* in predicted categories, identifying and evaluating key contributing mechanisms. This work provides a principled framework for selecting modeling strategies and offers practical guidance for building more accurate and robust forecasting systems. The code will be released upon publication.

## 1 Introduction

In many real-world prediction tasks, the precise temporal ordering of events is irrelevant. Instead, predicting the distribution of outcomes, where only the presence or absence of specific elements matters, is sufficient and often more practical.

For instance, in retail operations, probabilistic demand forecasting enables optimal inventory management and supply chain planning by modeling the full range of possible product demands without requiring sequence order (Nassibi et al., 2023; Larson, 2001). Similarly, in healthcare, clinical diagnosis systems treat disease categories as unordered sets within a single hospital admission. The presence of certain conditions is clinically more significant than the exact order in which they were diagnosed (Johnson et al., 2016; Mullenbach et al., 2018). Recommendation systems further exemplify this principle known as *basket prediction* (Rendle, 2020). Finally, many multi-label problems can naturally be framed as distribution forecasting tasks.

**The central focus of this paper** *is to model the future distribution of human actions over a fixed future horizon.* In this work we consider *Event Sequences* (EvS) (Osin et al., 2025; Udovichenko et al., 2024) - temporal records of human actions which underpin a wide range of decision-making systems across domains including healthcare (Johnson et al., 2016), financial transactions (Udovichenko et al., 2024; Mollaev et al., 2024; Yang & Xu, 2019), e-commerce (Li et al., 2021), recommender systems (Shevchenko et al., 2024; Klenitskiy et al., 2024; Zhelnin et al., 2025), and human action recognition (Surkov et al., 2024). Despite its practical importance and deceptively simple formulation, distribution forecasting for **EvS** remains significantly understudied.

Inspired by advances in Natural Language Processing (NLP), contemporary approaches to modeling sequential behavior often default to autoregressive generation predicting the next token conditioned on an exact prefix ordering (Karpukhin et al., 2024; Klenitskiy et al., 2024). While Next Token Prediction (NTP) has long dominated sequential modeling, Multi-Token Prediction (MTP) has recently gained traction due to its demonstrated improvements in model quality and generalization, particularly in tasks such as planning, code generation and EvS forecasting (Nagarajan et al., 2025; Bachmann & Nagarajan, 2024; Yu et al., 2025; Karpukhin & Savchenko, 2024).

**This raises a practical question:** When should we model future event distributions **explicitly**, and when is it worth preserving temporal structure through **implicit** order-preserving objectives methods like NTP or Multi-Token Prediction (MTP)? To answer this, we make the following contributions:

**(1) We systematically study the task of forecasting the distribution of future events** over a fixed horizon and demonstrate its importance as a viable alternative to autoregressive modeling in domains where the order of events is weakly informative or irrelevant. Our results show that this task is not only meaningful for practical applications but also enables simpler, more robust models that avoid pitfalls such as *mode collapse* (see Sec. 4.1).

**(2) Explicit vs. Implicit Objective Evaluation:** We conduct a rigorous empirical comparison of four training paradigms on seven public datasets: (1) Next Token Prediction , (2) Multi-Token Prediction with ordered output, (3) an order-invariant set prediction approach with post-hoc alignment, and (4) **GRU-Dist** - our proposed, explicit distribution forecasting objective. Our results demonstrate that explicitly modeling the future event distribution (GRU-Dist) consistently outperforms all order-preserving baselines across most domains. Surprisingly, when evaluated with order-invariant metrics, this superiority holds even on textual data, where sequential structure is traditionally assumed critical.

**(3) Connecting Dataset Structure to Model Performance:** We believe that the efficacy of sequential modeling is fundamentally governed by intrinsic dataset properties, since autoregressive paradigms developed for text we attempt to re-evaluate them accounting for dataset properties. We propose a following set of dataset characteristics and evaluations: the *Staticity Index (S)*, a distribution-based metric quantifying temporal drift across sequences; *Local Permutation Analysis*, which measures sensitivity to event shuffling within sliding windows; *Exponential Decay Parameter* $\lambda$ , capturing category imbalance and *Consecutive Repeat Rate (CRR)*, a measure to analyze ration of consecutively repetitive tokens, which are present in some real world e-commerce datasets as repetitive item clicks.

Our findings provide actionable guidance for informed model selection with respect to dataset properties, and demonstrate that next-token prediction is not universally optimal, even for large models across domains.

## 2 RELATED WORK

**Architectures for Event Sequences.** Modeling user actions sequentially by conditioning on past behavior has become an essential component of modern recommendation pipelines. These approaches effectively adapt ideas from natural language processing (NLP), particularly attention-based architectures (Kang & McAuley, 2018; Sun et al., 2019; Klenitskiy et al., 2024; Mezentsev et al., 2024). However, it remains unclear whether transformer-based architectures are indeed the most suitable for predicting future user actions. In *EBES* Osin et al. (2025) and in *Seq-NAS* Udovichenko et al. (2024), the authors demonstrate that RNN-based architectures outperform transformer-based models on **EvS** classification tasks. Delving deeper into this issue, Karpukhin & Savchenko (2025) investigate the limitations of transformers and proposes several modifications that enable them to surpass RNNs in classification performance. However, as the same work further reveals, these enhancements do not translate to improved performance in forecasting future tokens. In this work, we focus on RNN- and GPT-based architectures, as they remain the most applicable in this domain.

**Multi-Token vs. Single-Token Prediction.** Multi-Token Prediction (MTP) has recently gained traction due to its demonstrated improvements in model quality and generalization particularly in tasks such as planning, code generation (Nagarajan et al., 2025; Bachmann & Nagarajan, 2024; Yu et al., 2025). However, a key challenge lies in the common assumption that predicted tokens are conditionally independent Gloeckle et al. (2024).

*Teacherless Learning* Bachmann & Nagarajan (2024) offers an intermediate approach between Next-Token Prediction (NTP) and MTP, conceptually opposing teacher forcing. Unlike MTP, Teacherless Learning is grounded in a rigorous mathematical framework. While it does not accelerate inference, it addresses fundamental limitations of traditional NTP. As Nagarajan et al. (2025)

note: "Teacherless training and diffusion models comparatively excel in producing diverse and original output."

Although earlier work focused primarily on text generation, Karpukhin & Savchenko (2024) extended these ideas to **EvS**, demonstrating that multi-token generation and diffusion-based approaches indeed outperform the single-token paradigm. In this work, we investigate NTP, a multi-token strategy similar to that proposed in Karpukhin & Savchenko (2024) and propose a new explicit approach for distribution forecasting.

**Order Importance in EvS.** It has been established that permuting sequences in **EvS** datasets does not degrade performance on classification tasks (Osin et al., 2025; Moskvoretskii et al., 2024), an observation which significantly challenges the assumed sequential nature of this data type. Klenitskiy et al. (2024) investigates whether datasets from the domain of sequential recommender systems genuinely exhibit sequential structure. Specifically, the authors evaluate whether permuting sequences leads to performance degradation in next-token prediction tasks, and find that the extent of degradation varies by dataset, some datasets are more "sequential" than others. In this work, we extend this investigation beyond recommender systems and analyze local permutation invariance.

## 3 DATASETS

To evaluate the proposed methods and hypotheses, we conduct experiments on a diverse collection of real-world sequential datasets spanning multiple domains—including financial transactions, e-commerce, retail, music streaming, and literary text. A summary of key statistics is provided in Table 1; full descriptions, including preprocessing steps are available in Appendix A.3.

Table 1: Dataset statistics and characteristics.

| Dataset | ID | Domain | Sequences | Mean len | Target Field | Classes |
|---|---|---|---|---|---|---|
| Multimodal Banking Dataset 2024 | MBD | Transactions | 1.5M | 313 | Event type | 55 |
| AgeGroup Transactions | AGE | Transactions | 30K | 888 | Small group | 203 |
| X5 RetailHero | Retail | Retail | 40K | 112 | Level 2 | 43 |
| Alphabattle-2.0 | AB | Transactions | 1M | 213 | MCC category | 28 |
| Complete Works of Shakespeare | ShS | Text | 5K | 106 | Character | 65 |
| Megamarket (2024) | MM | E-commerce | 2.73M | 653 | Category ID | 9.8K |
| Zvuk (2024) | Zvuk | Music Streaming | 380K | 1020 | Artist ID | 210K |
| Taobao User Behavior | Taobao | E-commerce | 10K | 535 | Item category | 8K |

## 4 DATASET DIAGNOSTIC

### 4.1 TEMPORAL ORDER AND MODE COLLAPSE IN EVENT SEQUENCE MODELING

In time series and natural language modeling, precise temporal ordering is crucial. However, in domains like system logs or bank transactions, the *exact micro-temporal order* of events within short windows may be ambiguous or irrelevant—e.g., two unrelated log entries milliseconds apart could plausibly appear in either order without changing system semantics. We illustrate this effect in Appendix 4. This motivates a formal distinction between two types of temporal structure:

- **Local invariance**: Within a narrow window $W_t = (y_t, \ldots, y_{t+H})$, event order is semantically irrelevant—permutations of the same multiset are equally plausible.

- **Global structure**: Across broader time intervals, dependencies between consecutive windows remain meaningful; e.g., $p(W_2 \mid W_1)$ for $W_1 = (y_0, \ldots, y_{t-1})$ and $W_2 = (y_t, \ldots, y_{t+H})$ captures genuine temporal progression.

Conventional autoregressive (AR) models are trained to predict the next token $y_t$ given its full history $(y_0, \ldots, y_{t-1})$. To accommodate local invariance, one might relax this strict left-to-right dependency by defining a *prediction horizon* $\{y_t, \ldots, y_{t+H}\}$ and training the model to predict *any* event

within this window. Under the assumption of uniform uncertainty over the horizon, the training objective becomes:

$$\mathbb{E}_{k \sim \text{Uniform}[0,H]}\big[\log p(x = y_{t+k} \mid y_0, \dots, y_{t-1})\big] = \frac{1}{H+1} \sum_{m=0}^{H} \log p(x = y_{t+m} \mid y_0, \dots, y_{t-1}). \tag{1}$$

Critically, standard AR architectures use a *single output distribution* $q_t(\cdot)$ at time $t$ to score all tokens in the horizon. Under local permutation invariance, the optimal $q_t$ that maximizes the above objective is the empirical distribution over the multiset $\{y_t, \dots, y_{t+H}\}$. Consequently, the model learns a *static predictive distribution* over the entire window: $q_t \approx q_{t+1} \approx \cdots \approx q_{t+H}$. This static distribution becomes problematic at inference time. When generating sequences using deterministic decoding (e.g., argmax or low-temperature sampling), the model outputs:

$$\hat{y}_{t+k} = \arg \max_x q_{t+k}(x) \approx \arg \max_x q_t(x), \quad \forall k \in [0, H].$$

Since $q_t$ is dominated by the most frequent event in the window, the model repeatedly predicts the *empirical mode* of $\mathcal{W}_t$, suppressing rarer—but valid—events. We term this phenomenon *temporal mode collapse*.

We propose that explicitly modeling the distribution of events across entire windows, rather than enforcing pointwise predictions, offers a principled resolution. This allows models to better capture the stochastic nature of real-world event sequences while avoiding degenerate solutions.

## 4.2 STATICITY INDEX

Before fitting neural models, we quantify how the event distribution of each sequence changes over time. Previous studies have shown that drift of the temporal distribution can strongly influence forecast performance, including context-driven shift (Chen et al., 2024), seasonality-induced shifts (Liao et al., 2025), and temporal dataset shift benchmarks (Yao et al., 2022). Motivated by these findings, we measure the stability or dynamic of the empirical event distribution within each dataset. To this end, we plot the *Shape* score drift for every dataset to reveal whether their event distributions remain nearly static or show meaningful temporal variation.

Several datasets contain sequences with nearly static behavior; to verify this, we plot the *Shape* score drift for each dataset.

### 4.2.1 PER-FEATURE DISSIMILARITY SCORE

**Procedure.** For each sequence, we fix a window length $W$ and stride $s$, then slide the window across the timeline. At every position $i$, we extract the feature distribution $P_i$ within the current window and compare it with the baseline distribution $P_0$ computed from the first window. To compare them, we suggest to leverage the following score:

Let $P_0$ and $P_i$ denote the empirical distributions in the reference window and the $i$-th window, respectively.

**Discrete features.** For categorical attributes defined on $\mathcal{A}$ we employ the *total variation (TV) distance*, $\text{TV}(P_0, P_i) = \frac{1}{2} \sum_{a \in \mathcal{A}} \big|P_0(a) - P_i(a)\big|$. Because lower TV indicates higher similarity, we report its complement $(1 - \text{TV})$, so that higher values consistently reflect better alignment.

**Continuous features.** For numerical attributes we use the *Kolmogorov–Smirnov* statistic. Let $F_0$ and $F_i$ be the empirical CDFs corresponding to $P_0$ and $P_i$. The KS divergence is $\text{KS}(P_0, P_i) = \sup_{x \in \mathbb{R}} \big|F_0(x) - F_i(x)\big|$. Analogously, we report the similarity score $1 - \text{KS}$.

**Shape score.** For window $i$ we propose to compute each feature's distance using the appropriate formula above and then average across all features: $\text{Shape}(P_0, P_i) = \frac{1}{M} \sum_{j=1}^{M} d_j\big(P_0^{(j)}, P_i^{(j)}\big)$, where $M$ is the number of features and $d_j$ is TV when the $j$th feature is categorical, and KS otherwise. Plotting $i \mapsto \text{shape}(P_0, P_i)$ yields the drift curves used throughout this paper.

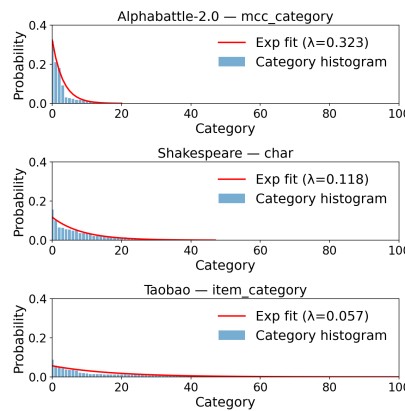

Figure 1: Distribution of categories in datasets. We present normalized number of categories.

Table 2: Dataset statistics: exponential decay parameter ($\lambda$), Consecutive Repeat Rate (CRR), total number of distinct categories (TCD), Staticity index ($S$; ==average distributional similarity over time, 1 = fully stationary==), and perplexity (PPL) increase after full shuffle.

| Dataset | $\lambda$ | CRR | TCD | S | PPL |
|---|---|---|---|---|---|
| *Banking domain* | | | | | |
| MBD | 0.415 | 1.718 | 55 | 0.842 | 1.02× |
| AB | 0.305 | 1.518 | 28 | 0.725 | 1.08× |
| Age | 0.245 | 1.148 | 203 | 0.772 | 1.24× |
| Retail | 0.185 | 1.372 | 43 | 0.782 | 1.27× |
| *Text* | | | | | |
| ShS | 0.118 | 1.019 | 64 | 0.803 | 5.09× |
| *Recommender Systems* | | | | | |
| Taobao | 0.016 | 4.492 | 1.9K | 0.650 | 13.00× |
| MM | 0.005 | 3.502 | 9.8K | 0.406 | 10.87× |
| Zvuk | 0.003 | 1.239 | 210K | 0.363 | 5.05× |

With these definitions we shift the window across the entire sequence and plot trajectory $i \mapsto \mathrm{shape}(P_0, P_i)$, obtaining time-resolved drift curve that summarises how the distribution evolves over the time.

### 4.2.2 STATICITY IN DATASETS

Across banking datasets (MBD, Retail, Age, AlphaBattle) the majority of user sequences form static clusters with negligible temporal drift (Figure 5, Appendix A.4). In contrast, RecSys data such as ZVUK exhibit more diverse and volatile trajectories (Figure 6, Appendix A.4), while the Shakespeare dataset, despite being textual, resembles banking data with largely flat drift patterns (Figure 7, Appendix A.4). Detailed analyses for individual datasets are provided in the Appendix A.4.

**Motivation.** These observations motivate a prevent-level *staticity index* that can be computed *before* model training to guide the choice of modeling strategy. Unlike the single–anchor variant (first window vs. all others), we adopt a more robust, multi–anchor formulation.

**Staticity index.** Fix a window length $W$ and stride $s$. For each sequence $u$ with per–window distributions $\{P_i^{(u)}\}_{i=1}^{I_u}$, choose anchors $\mathcal{R}_u$ (uniformly at random, $R = 3$). The per–sequence score is the average shape–similarity

$$S^{(u)} = \frac{1}{RI_u} \sum_{r \in \mathcal{R}_u} \sum_{i=1}^{I_u} \mathrm{Shape}\left(P_r^{(u)}, P_i^{(u)}\right),$$

and the dataset–level index is $\mathrm{Staticity} = \frac{1}{N} \sum_{u=1}^{N} S^{(u)}$.

Thus, the staticity index quantifies the temporal stability of a sequence's multi-feature distribution: higher values (close to 1) reflect stronger staticity (quasi-stationarity), whereas values near zero indicate pronounced drift. Importantly, the conclusions derived from the computed staticity index (Table 2) align with those previously inferred from the qualitative analysis of the plots.

### 4.3 LOCAL PERMUTATION OF EVENTS

==To assess the importance of temporal order, we apply a local permutation operator parameterized by a window radius $w$. For each position $i$, we construct a symmetric window $[i - w, i + w]$, and the event at position $i$ is allowed to move only within this window. Specifically, we construct a square cost matrix filled with random values, mask out entries corresponding to positions outside the window (and all padding tokens), and compute the constrained permutation via the Hungarian algorithm (Kuhn, 1955). We use $w \in \{0, 1, 4, 16, -1\}$, where larger values correspond to stronger==

disruption of local order. The case $w = -1$ removes the positional constraint completely and allows a global permutation of the sequence.

Importantly, even when $w = -1$, we strictly prevent mixing between the historical part of the sequence and the target part: the two segments are permuted independently. This ensures that the model never sees target tokens reintroduced into the history during shuffling. For each window size $w$, we train and evaluate the model under the corresponding level of local permutation, enabling us to study how different datasets respond to disrupted temporal structure.

## 4.4 OTHER STATISTICS

We also report the exponential decay parameter $\lambda$, which quantifies how quickly category frequencies decline in each dataset. Specifically, $\lambda$ is the rate parameter of an exponential distribution fitted to the empirical histogram of event categories. This fit provides a compact measure of distributional imbalance: larger $\lambda$ values indicate a steeper decay and, consequently, a stronger dominance of the most frequent categories. Figure 1 illustrates the fitted exponential curves alongside the empirical histograms for several datasets. The corresponding $\lambda$ values for all datasets are reported in Table 2.

Additionally, we report the Consecutive Repeat Rate (CRR)—the average length of uninterrupted runs of identical tokens. Higher CRR indicates more repetition, which can inflate short-term prediction accuracy. CRR values are listed in Table 2.

## 5 DISTRIBUTION FORECASTING METHODS

We study the task of forecasting a distribution of a sequence over some horizon $N$ given its history. To this end, we consider several training objectives — autoregressive, target-based, matched, and our order-invariant formulation. For all experiments $N$ is fixed as 32.

### 5.1 AUTOREGRESSIVE LOSS

Let $x_{1:T}$ be a sequence with $x_t \in \{1, \ldots, K\}$. The model parameterises conditional next–event probabilities $p_\theta(x_{t+1} \mid x_{1:t})$ given the preceding context $x_{1:t}$. The sequence log–likelihood factorises as: $\log p_\theta(x_{1:T}) = \sum_{t=1}^{T} \log p_\theta(x_{t+1} \mid x_{1:t})$.

### 5.2 TARGET LOSS

In this setting the model predicts an entire block of $L$ future events *in a single forward pass*, using a fixed prefix $x_{1:T}$ as context; no teacher forcing is applied inside the horizon. Let $\hat{p}_{T+1}, \ldots, \hat{p}_{T+L}$ be the categorical distributions produced for positions $T+1$ through $T+L$. The target loss is the sum of negative log-likelihoods for that block: $\mathcal{L}_{\text{target}}^{(L)} = \sum_{i=T+1}^{T+L} -\log \hat{p}_i(x_i \mid x_{1:T})$

Unlike the autoregressive objective, every term is conditioned on *the same* prefix $x_{1:T}$; the model **GRU-Target** therefore learns to produce an entire horizon coherently without receiving the ground-truth events $x_{T+1:T+L-1}$ as intermediate inputs.

### 5.3 MATCHED LOSS

When the temporal order of future events is weakly informative, forcing the model to predict both the *events* and their *exact positions* needlessly penalises near-correct outputs. The **GRU-Matched** model adapts the matching idea of Karpukhin & Savchenko (2024), aligning each target event with the nearest prediction within a tolerance window of size $m$, treated as a hyperparameter.

Let a fixed prefix $x_{1:T}$ condition a one-shot block prediction $\hat{p}_{T+1:T+L}$; let $x_{T+1:T+L}$ be the corresponding ground truth. With a permutation $\sigma$ constrained by $|\sigma(i) - i| \leq m$, the matched loss is
$$\mathcal{L}_{\text{match}}^{(m)} = \min_{\substack{\sigma \in \mathcal{A} \\ |\sigma(i)-i| \leq m}} \sum_{i=T+1}^{T+L} -\log \hat{p}_{\sigma(i)}(x_i \mid x_{1:T}).$$

At $m = 0$ it reduces to plain block cross-entropy; as $m$ grows, the objective becomes progressively order-invariant. The minimisation is solved with the Hungarian algorithm on the cost matrix $\ell_{ij} = -\log \hat{p}_j(x_i \mid x_{1:T})$.

### 5.4 Order-Invariant Distribution Parameterization

When the order of future events is not informative, it is sufficient to model only the *event type distribution* rather than their precise temporal arrangement. We therefore introduce the **GRU-Dist** model, which represents each sequence as a *bag of events* and is trained to match the empirical distribution.

Let $H_t = \{x_1, \ldots, x_t\}$ be the multiset of events observed so far. A neural encoder $f_\theta$ maps $H_t$ to logits, which are converted to probabilities $\boldsymbol{\pi}_t = \mathrm{softmax}\big(f_\theta(H_t)\big) \; \in \; \Delta^{K-1}$, where $\Delta^{K-1}$ is the probability simplex in $\mathbb{R}^K$. For a sequence of length $L$ we form its empirical distribution $\hat{p}_k = \frac{1}{L} \sum_{t=1}^{L} \mathbf{1}\{x_t = k\}$, and minimize $\ell(\theta) = \mathrm{D}_{\mathrm{KL}}\big(\hat{p} \,\|\, \boldsymbol{\pi}(\theta)\big)$.

Unlike autoregressive objectives that require $L \times K$ logits per sequence, our order-invariant head outputs only a single $K$-dimensional vector. This reduces both computational and memory costs by a factor of $L$, while remaining well suited for datasets where event order carries little information.

## 6 Evaluation

For each configuration $Dataset \times Method \times LocalShuffle$ we perform an extensive hyperparameter optimization of 100 trails, technical details are given in Appendix A.1.

### 6.1 Baselines

**We consider four simple baselines. (1) Ground Truth** uses the original sequences as a sanity check and reference point for metrics such as Cardinality. **Repeat** extends a sequence by copying its most recent observations into the forecast horizon of the lenght $N$. **Mode** outputs the users most frequent category for all $N$, illustrating the tendency of autoregressive models to collapse into trivial mode repetition—a behavior that may be overestimated by order-dependent metrics (e.g., Accuracy, Levenshtein distance). Finally, **HistSampler** generates sequences by sampling from the empirical histogram of past users sequence, thereby preserving marginal category frequencies while discarding temporal dependencies.

### 6.2 Neural Backbones

We evaluate two neural backbone architectures for sequence modeling:

- **GRU:** The standard Gated Recurrent Unit (GRU) Cho et al. (2014) excels in capturing local dependencies and stationary patterns in short to moderately long time series.
- **GPT:** GPT-2 Radford et al. (2019), a causal Transformer-based model capable of modeling long-range dependencies, crucial for sequences with complex contextual interactions and implicit event relationships.

### 6.3 Multi-Token Prediction via Sampling

**Sampling in the order-sensitive models:** Autoregressive decoding with greedy argmax often collapses to the modal category. A simple remedy is to *sample* from the predictive categorical distribution instead of always taking the maximum, which reduces *mode collapse* and improves order-invariant metrics. For autoregressive and block-prediction models this sampling is straighforward, as logits at each step define the distribution, in our order-invariant method the distribution itself is parameterized directly, making sampling the natural decoding mechanism. We did not analyze more sophisticated sampling approaches such as beam search and our preliminary experiments with temperature sampling did not provide stable improveent across datasets, so we do not use them.

**Sampling in the order-invariant model:** Given a predicted categorical distribution $\pi = (\pi_1, \ldots, \pi_K)$ and a target length $L$, we first compute the expected fractional counts $\hat{n}_k = L\pi_k$.

Table 3: Next $N = 32$ tokens forecasting. *Matched-F1 (micro)* for all datasets and methods including baselines. [†] denotes sampled version of method.

| Method | MBD | Age | AB | Retail | ShS | Taobao | MM | Zvuk |
|---|---|---|---|---|---|---|---|---|
| GT | 1.000 | 1.000 | 1.000 | 1.000 | 1.000 | 0.926 | 1.000 | 1.000 |
| Mode | 0.520 | 0.331 | 0.380 | 0.219 | 0.158 | 0.117 | 0.156 | 0.113 |
| Repeat | 0.830 | 0.680 | 0.700 | 0.661 | 0.587 | **0.257** | **0.318** | **0.274** |
| HistSampler | 0.804 | 0.632 | 0.680 | 0.640 | 0.533 | 0.197 | 0.244 | 0.226 |
| GRU | 0.528 | 0.477 | 0.375 | 0.207 | 0.596 | 0.222 | 0.250 | 0.148 |
| GRU[†] | 0.771 | 0.628 | 0.641 | 0.609 | 0.596 | 0.146 | 0.171 | 0.126 |
| GPT | 0.524 | 0.476 | 0.373 | 0.212 | 0.594 | 0.223 | 0.250 | 0,192 |
| GPT[†] | 0.776 | 0.627 | 0.629 | 0.611 | 0.603 | 0.151 | 0.188 | 0,174 |
| GRU-Target | 0.541 | 0.370 | 0.403 | 0.398 | 0.299 | 0.196 | 0.267 | 0.143 |
| GRU-Target[†] | 0.808 | 0.633 | 0.670 | 0.641 | 0.572 | 0.154 | 0.201 | 0.140 |
| GRU-Matched | 0.847 | 0.704 | 0.676 | 0.708 | 0.688 | 0.203 | 0.272 | 0.202 |
| GRU-Matched[†] | 0.827 | 0.653 | 0.647 | 0.667 | 0.634 | 0.155 | 0.203 | 0.134 |
| GRU-Dist | **0.856** | **0.725** | **0.736** | **0.719** | **0.705** | 0.178 | 0.247 | 0.239 |

Since these values are not integers, we obtain discrete category counts $(n_1, \ldots, n_K)$ using Hamilton's method Balinski & Young (2010), $(n_1, \ldots, n_K) = \mathrm{Hamilton}(\hat{n}_1, \ldots, \hat{n}_K), \sum_{k=1}^{K} n_k = L$. This method distributes $L$ discrete slots among categories in proportion to their predicted probabilities $\pi_k$ and ensures that the total count equals $L$.

## 6.4 Metrics

Many classical sequence metrics (e.g., *Accuracy*, *Levenshtein distance*, *F1-score*) are defined with respect to a fixed token order and therefore penalize any permutation of events, even when such reordering is irrelevant for the problem at hand. To overcome this limitation, we introduce an order-invariant *Matched-F1* score, which treats sequences as *bags of events*.

To avoid order dependence we redefine true-positive, false-positive and false-negative terms. Let $g_k$ and $\hat{g}_k$ denote the ground-truth and predicted multiplicities of class $k$ in the window. We set

$$(\mathrm{TP}_k, \mathrm{FP}_k, \mathrm{FN}_k) = \big(\min(g_k, \hat{g}_k),\ \max(0, \hat{g}_k - g_k),\ \max(0, g_k - \hat{g}_k)\big).$$

Based on this definitions, we compute our *Matched-F1* with **micro-** and **macro-**averaging, analogous to the conventional *F1-score* formulation. Detailed definition of this metric placed in Appendix A.6.1

To assess diversity, we use **Cardinality** (see Appendix A.6.2), which measures the number of distinct categories generated by the model. Low values signal *mode collapse*, while values close to the ground-truth indicate faithful event variety.

For completeness, we also report **Levenshtein distance**, an order-sensitive metric that, although less relevant to our setting, provides a complementary reference for order preserving methods (Table 4).

## 7 Results

**Dataset-level statistics.** The staticity index serve as useful diagnostics for anticipating whether sequence order is relevant. Results are presented in Table 2. In banking datasets, a single modal category dominates—accounting for more than 50% of all events—leading to high values of both $\lambda$ and the staticity index. This dominance is also associated with a pronounced performance drop under local permutations, suggesting limited reliance on sequential order.

**Local permutation experiments** (see Section 4.3) further corroborate these findings; results are shown in Figure 3. Shakespeare and Zvuk exhibit sharp performance degradation when sequences

Table 4: Next $N = 32$ tokens forecasting. *Levenshtein* for all datasets and methods including baselines. † denotes sampled version of method.

| Method | MBD | Age | AB | Retail | ShS | Taobao | MM | Zvuk |
|---|---|---|---|---|---|---|---|---|
| GT | 1.000 | 1.000 | 1.000 | 1.000 | 1.000 | 0.926 | 1.000 | 1.000 |
| Mode | 0.520 | 0.331 | 0.378 | 0.219 | 0.158 | 0.117 | 0.156 | 0.113 |
| Repeat | 0.516 | 0.310 | 0.253 | 0.229 | 0.150 | 0.118 | 0.162 | 0.101 |
| GRU | 0.520 | **0.390** | 0.374 | 0.194 | 0.200 | 0.222 | **0.250** | 0.139 |
| GRU† | 0.491 | 0.292 | 0.316 | 0.211 | 0.166 | 0.104 | 0.125 | 0.067 |
| GPT | 0.520 | **0.390** | 0.371 | 0.195 | **0.216** | 0.223 | **0.250** | **0.167** |
| GPT† | 0.490 | 0.291 | 0.310 | 0.212 | 0.174 | 0.111 | 0.136 | 0.092 |
| GRU-Target | **0.527** | 0.344 | **0.385** | **0.256** | 0.192 | 0.175 | 0.236 | 0.119 |
| GRU-Target† | 0.507 | 0.275 | 0.322 | 0.223 | 0.134 | 0.083 | 0.122 | 0.053 |
| GRU-Matched | 0.519 | 0.309 | 0.332 | 0.245 | 0.138 | 0.162 | 0.230 | 0.105 |
| GRU-Matched† | 0.492 | 0.257 | 0.287 | 0.216 | 0.122 | 0.087 | 0.127 | 0.051 |
| GRU-Dist | 0.421 | 0.206 | 0.257 | 0.161 | 0.096 | 0.072 | 0.116 | 0.062 |

are shuffled, indicating strong local sequential structure. In contrast, most banking datasets show little to no degradation, reflecting the irrelevance of event order. This trend is especially evident in Figure 2, which illustrates minimal perplexity degradation under shuffling for these datasets.

**Matched-F1 performance.** Order-invariant methods achieve the best overall performance on most datasets, significantly outperforming order-sensitive approaches (Table 3). *GRU-Dist* consistently outperforms *GRU-Matched*. Exceptions are Taobao and Megamarket, where *GRU-Dist* underperforms. These datasets exhibit a high Consecutive Repeat Rate (CRR, Table 2), and other models exploit this by repeating recent categories. *GRU-Dist*, by design, cannot leverage such local repetition. All learning methods struggle on Taobao, Megamarket, and Zvuk due to their very low exponential decay parameter $\lambda$ and extremely high cardinality (Table 2). Here, *Repeat* baseline performs best. Sampling improves Matched-F1 for most order-sensitive models by alleviating *mode collapse*, but not on Taobao, Megamarket, and Zvuk (again due to low $\lambda$ and high cardinality). Even with sampling, they remain inferior to order-invariant models.

**Levenshtein performance.** The *Mode* baseline is strong compared to order-sensitive methods in MBD, Age, AlphaBattle and Retail, highlighting the difficulty of modeling precise order (Table 4). As expected, order-agnostic models perform worse, since they impose no ordering constraints. Surprisingly, NTP models remain competitive on Taobao and Megamarket despite severe *mode collapse* (cardinality = 1): their local-mode predictions adapt better to fast distributional shifts than the static global mode, consistent with their low Staticity index $S$ (Table 2).

# 8 CONCLUSION

Our study demonstrates that model performance in event-sequence forecasting depends strongly on dataset properties and on whether order-invariant or order-sensitive evaluation is appropriate.

When temporal order is largely irrelevant, order-sensitive methods suffer from *mode collapse*, performing similarly to the *Mode* baseline. In this case, order-invariant metrics are more appropriate. Under these metrics, *GRU-Dist* is generally the best, except when the category distribution is highly skewed (low $\lambda$) or the repetition is high (high CRR), where the *Repeat* baseline dominates.

When temporal structure is strong (low Staticity index $S$, high Consecutive Repeat Rate (CRR), significant Perplexity increase under local permutation), order-sensitive metrics become more appropriate, and autoregressive models are preferable, often outperforming other baselines even in the presence of severe *mode collapse*.

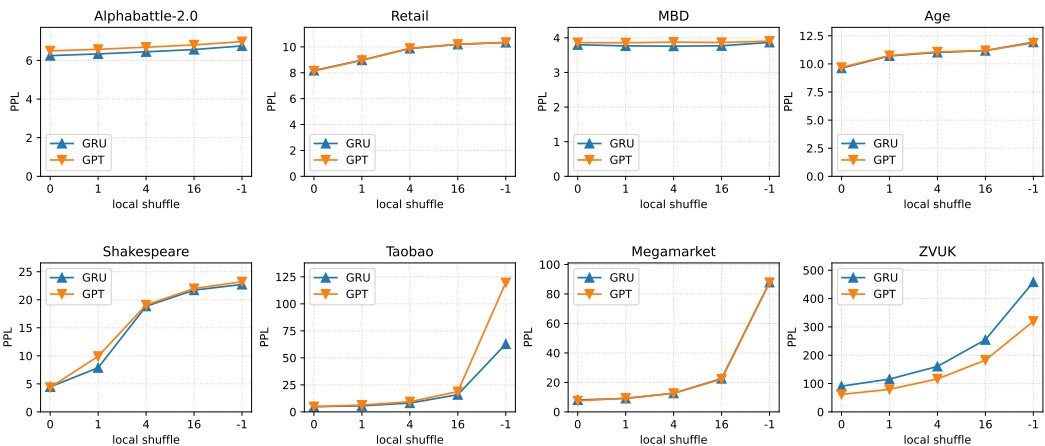

Figure 2: Next $N = 32$ tokens forecasting. Perplexity results.

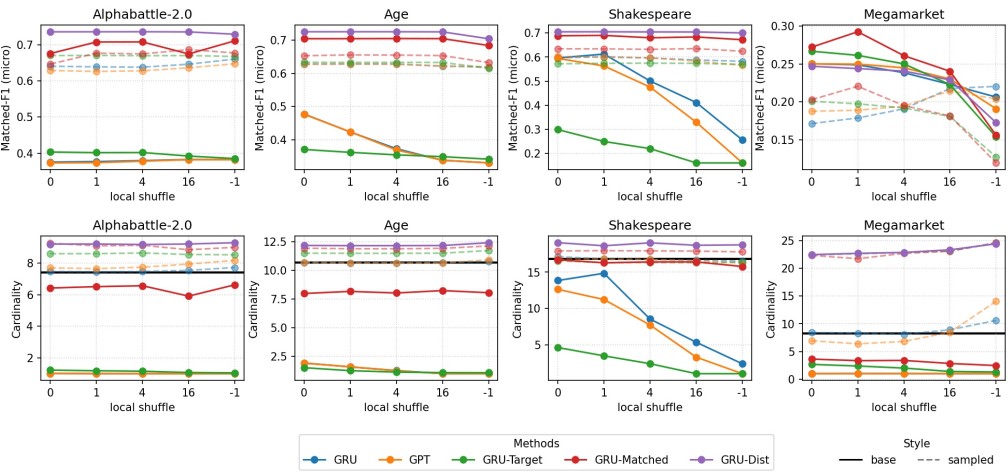

Figure 3: Effect of Local Event Shuffling on Model Performance. We report Matched-F1 score and Carnality for four datasets. Results for other datasets and metrics can be found in Appendix A.8

Cardinality also proves to be a useful diagnostic of *mode collapse*: in datasets like Shakespeare, shuffling removes structural cues and autoregressive models degenerate to the modal category. More broadly, when no meaningful local ordering exists, models tend to collapse to the mode (Figure 3).

Taken together, these results highlight the value of simple dataset-level diagnostics for anticipating model behavior, and demonstrate the advantages of order-invariant objectives in domains such as retail and banking, where event presence matters more than sequence order.

Indeed, it is worth noting that the proposed *GRU-Dist* method can be extended from single-category forecasting to multi-feature prediction through cascade modeling.

**Acknowledgment on LLM assisted writing:** This paper used open access Qwen3-Max, in some parts of the paper, for proofreading and text rephrasing in accordance with formal style.

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

# A APPENDIX

## A.1 HPO DETAILS

For hyperparameter optimization (HPO), we use Optuna (Akiba et al., 2019) with the Tree-structured Parzen Estimator (TPE) sampler. For each model–dataset pair, we allocate an HPO budget of 100 training runs, capping the total computational cost at 18 NVIDIA A100 GPU-days. We reserve 15% of the training set as a validation subset for early stopping and hyperparameter selection. The best-performing hyperparameters are then used to train the final model for evaluation and all subsequent study experiments.

## A.2 LOCAL GLOBAL TEMPORAL INVARIANCE

In Figure 4 we illustrate local / global invariance.

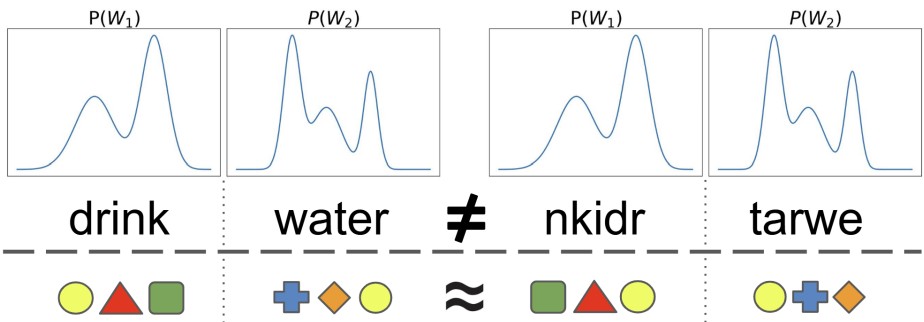

Figure 4: Example how order importance differs in different types of data. Even though in both cases horizon distribution doesnt change, event sequence still make sence after permut inside intervals.

## A.3 DATASETS DESCRIPTION AND PREPROCESSING

**MBD** [1] is a multimodal banking dataset introduced in Mollaev et al. (2024). The dataset contains an industrial-scale number of sequences, with data from more than 1.5 million clients in 2 year period. Each client corresponds to a sequence of events. This multi-modal dataset includes card transactions, geo-position events, and embeddings of dialogs with technical support. For our analysis, we use only card transactions. We use a temporal train–test split: transactions from the first year form the training set, and those from the second year form the test set.

**Age** dataset[2] consists of 44M anonymized credit card transactions representing 50K individuals. The target is to predict the age group of a cardholder that made the transactions. Each transaction includes the date, type, and amount being charged. The dataset was first introduced in scientific literature in work Babaev et al. (2022). We perform a user-based split: 80% of sequences are assigned to the training set, and the remaining 20% of sequences are held out for testing.

**Retail** dataset[3] comprises 45.8M retail purchases from 400K clients, with the aim of predicting a client's age group based on their purchase history. Each purchase record includes details such as time, item category, the cose, and loyalty program points received. The age group information is available for all clients, and the distribution of these groups is balanced across the dataset. The dataset was first introduced in scientific literature in work Babaev et al. (2022). We perform a user-based split: 80% of sequences are assigned to the training set, and the remaining 20% of sequences are held out for testing.

---

[1] https://huggingface.co/datasets/ai-lab/MBD
[2] https://ods.ai/competitions/sberbank-sirius-lesson
[3] https://ods.ai/competitions/x5-retailhero-uplift-modeling

**Alphabattle-2.0** datase [4] The AlfaBattle 2.0 dataset contains bank customers' transaction records over two years, with the goal of predicting loan default based on behavioral history. Each record includes 18 features (3 numeric, 15 categorical) per transaction. We use the official test split provided by the dataset creators.

**Shakespeare** Dataset consists of character-level text extracted from Shakespeare's works, preprocessed into individual speech segments. Each speech is tokenized using a vocabulary of unique characters mapped to integer codes based on frequency. The final dataset is split into train and test sets (80/20). The dataset is designed for character-level language modeling and was selected due to it obvious temporal importance.

**Zvuk** dataset[5] is introduced in 2024 and contains 244.7M music listening events grouped into 12.6M sessions from 382K users, recorded during the same five-month period (January–May 2023). In total, it spans 1.5M unique tracks. Each record includes a user ID, session ID, track ID, timestamp, and play duration (considering only plays covering at least 30% of track length). The dataset is tailored to music consumption, excluding podcasts and audiobooks, and enables evaluation of recommendation models in domains with stronger sequential dynamics. We use a temporal train–test split: transactions from the first two months form the training set, and other two month form the test set.

**MegaMarket** dataset[6] is introduced in 2024 and comprises 196.6M user interactions collected over a five-month period (January–May 2023). It covers 2.7M users, 3.56M items, and 10,001 product categories, with events including views, favorites, cart additions, and purchases. Each record contains a user ID, item ID, event type, category ID, timestamp, and normalized price. The dataset represents large-scale e-commerce behavior and is intended for sequential recommendation tasks. This dataset follows the same temporal train/test split as Zvuk.

**Taobao** [7] The dataset comprises user behaviors from Taobao, including clicks, purchases, adding items to the shopping cart, and favoriting items. These events were collected between November 18 and December 15. The training set encompasses data from November 18 to December 1, while the test set includes clicks from December 2 to December 15.

A.4  STATICITY INDEX PLOTS FOR KEY DATASETS

For each dataset, we compute drift trajectories for all sequence and cluster them into a small number of groups with internally consistent dynamics (Figure 5–7). Across banking datasets (MBD, Retail, Age, Alphabattle) the dominant clusters are static, as exemplified for **MBD** (Figure 5c), these clusters exhibit negligible temporal drift. For such sequences, learning the user's category distribution suffices to forecast the next block of events. Trajectories with pronounced drift are rare. In MBD specifically, such sequences are observed in fewer than 6% of users (Figure 5b).

In contrast to banking datasets, recommender–system data exhibit much greater variability. In **ZVUK** (Figure 6), two characteristic regimes dominate: one cluster shows a sharp initial drop from the baseline followed by persistent high-variance fluctuations, while another appears quasi-static yet remains noisy around its trend. Such patterns reflect the broader nature of recommender logs: users interact with a large and diverse sets of items, and their behavior shifts more frequently than in retail domains where event types are limited and highly regular. And as a consequence, their later-window distributions are more clearly separated from the first-window distribution.

The outlier in this collection is the **Shakespeare** text dataset (Figure 7). Although it is non-transactional, its dynamics resemble banking data more than recommender logs: drift trajectories are mostly flat and volatility remains low. At the same time, weak periodic or gradual shifts are observable, indicating that the sequences are not fully static but display a modest degree of temporal variation.

---

[4] https://www.kaggle.com/datasets/mrmorj/alfabattle-20
[5] https://www.kaggle.com/datasets/alexxl/zvuk-dataset
[6] https://www.kaggle.com/datasets/alexxl/megamarket?select=megamarket.parquet
[7] https://tianchi.aliyun.com/dataset/46

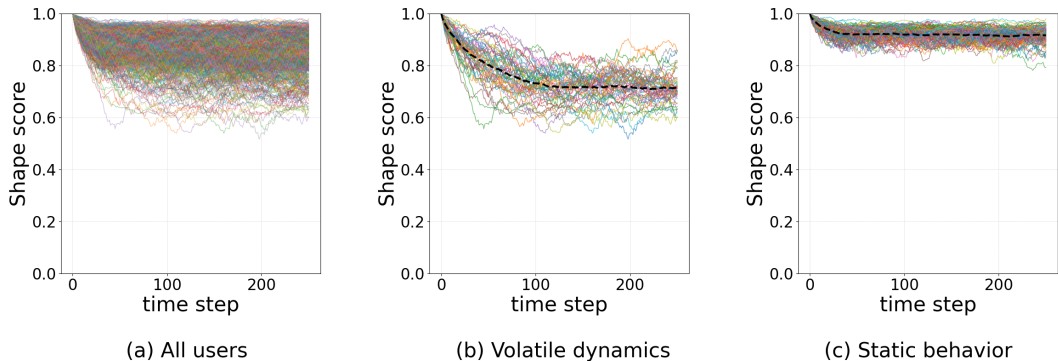

(a) All users      (b) Volatile dynamics      (c) Static behavior

Figure 5: Shape score drift for MBD dataset

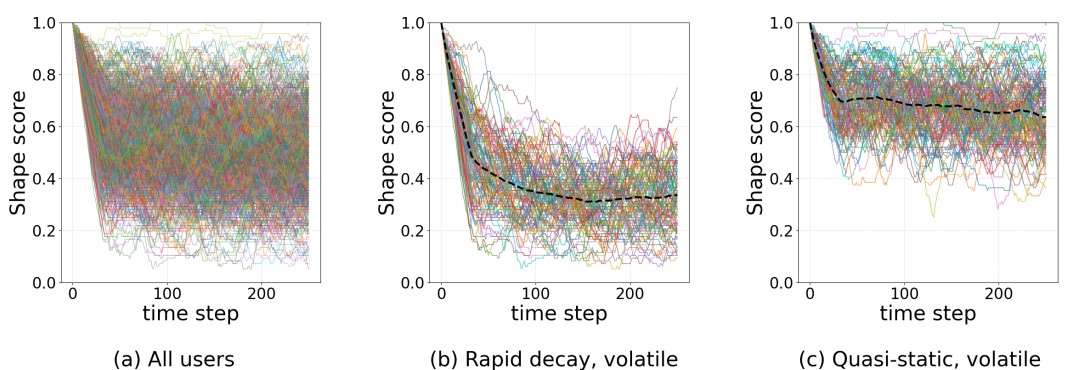

(a) All users      (b) Rapid decay, volatile      (c) Quasi-static, volatile

Figure 6: Shape score drift for ZVUK dataset

## A.5   Features Impact in Category Forecasting Quality

We investigated whether predicting a target feature benefits more from incorporating the full feature vector or from relying exclusively on its own historical values.

On the MBD dataset, experiments in the *All-to-One* and *One-to-One* modes reveal that the autoregressive model's performance degrades when exposed to complete with the complete feature vector. The additional inputs act as noise, impeding the model's ability to reproduce the mode of the target distribution. In the *One-to-One* mode—where the model sees only the history of the target feature—it easily learns the mode and reports a formal increase in accuracy; however, this gain is illusory, as the generated sequences become overly uniform and lack realism 5.

Table 5: Effect of training with all tokens vs. event type only (*Matched-F1 micro*).

| Dataset | Change (%) |
|---------|-----------|
| MBD | $+2.85$ |
| AGE | $-24.94$ |
| MM | $+13.66$ |

By contrast, on datasets with a strong sequential structure, such as *Megamarket*, the opposite pattern emerges. The autoregressive mechanism leverages ordering information and, when augmented with additional features, predicts beyond mere modal values, resulting in a significant improvement in performance metrics.

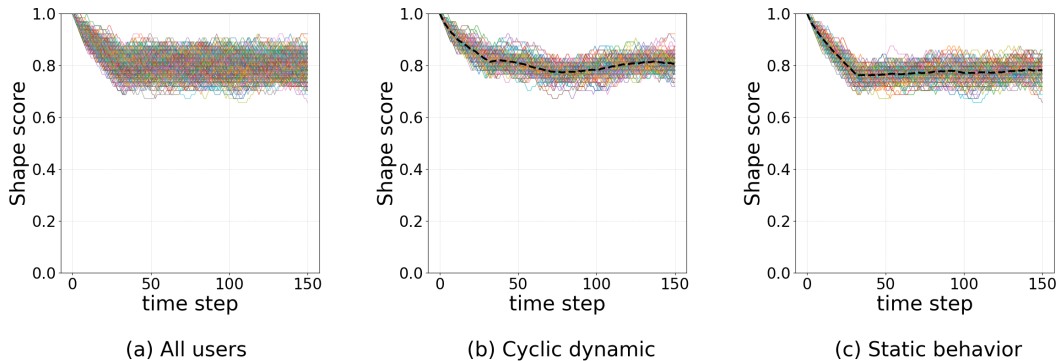

Figure 7: Shape score drift for Shakespeare dataset

## A.6 METRICS

### A.6.1 MATCHED-F1 MICRO

**Precision and recall.**

$$\text{Prec}_k = \frac{\text{TP}_k}{\text{TP}_k + \text{FP}_k}, \qquad \text{Rec}_k = \frac{\text{TP}_k}{\text{TP}_k + \text{FN}_k}.$$

**Macro averaging.**

$$F1_{\text{macro}} = \frac{1}{K} \sum_{k=1}^{K} \frac{2\,\text{Prec}_k\,\text{Rec}_k}{\text{Prec}_k + \text{Rec}_k}.$$

Each class contributes equally; the score is sensitive to rare categories.

**Micro averaging.** Aggregating counts over classes,

$$\text{TP} = \sum_k \text{TP}_k, \qquad \text{FP} = \sum_k \text{FP}_k, \qquad \text{FN} = \sum_k \text{FN}_k, \qquad (2)$$

$$F1_{\text{micro}} = \frac{2\,\text{TP}}{2\,\text{TP} + \text{FP} + \text{FN}}. \qquad (3)$$

This variant weights categories by frequency and reflects overall throughput.

### A.6.2 CARDINALITY METRIC.

Let $G_i = \left(x_{t+1}^{(i)}, \ldots, x_{t+L}^{(i)}\right)$ denote the $L$-step segment generated for sequence $i$ and $\mathcal{C}(G_i) = \{x \in G_i\}$ the set of *distinct* categories appearing in that segment. We define the per-sequence cardinality as

$$C_i \;=\; \big|\mathcal{C}(G_i)\big|.$$

The dataset-level score is the average

$$\text{Cardinality} \;=\; \frac{1}{N} \sum_{i=1}^{N} C_i,$$

where $N$ is the number of sequences under evaluation. An *overall* variant first concatenates all generated segments, $\tilde{G} = \bigcup_i G_i$, and reports $C_{\text{overall}} = |\mathcal{C}(\tilde{G})|$.

**Purpose.** Cardinality captures the *category diversity* produced by a model: low values signal *mode collapse*, whereas values close to the ground-truth cardinality indicate faithful reproduction of event variety. We compute the metric for both generated ($C_{\text{gen}}$) and reference ($C_{\text{orig}}$) sequences, allowing direct comparison of a model's diversity against empirical data.

Table 6: Comparison of autoregressive baselines under the original one-to-one setup and the all-to-all variant. Metrics are computed only on the target event category.

| Metric | MBD | AGE | AB | Retail | ShS | Taobao | MM | Zvuk |
|---|---|---|---|---|---|---|---|---|
| Matched F1 (one-to-one) | 0.528 | 0.476 | 0.375 | 0.208 | 0.596 | 0.222 | 0.250 | 0.148 |
| Matched F1 (all-to-all) | 0.440 | 0.474 | 0.375 | 0.006 | 0.612 | 0.176 | 0.026 | 0.004 |
| Levenshtein (one-to-one) | 0.520 | 0.390 | 0.374 | 0.194 | 0.200 | 0.222 | 0.250 | 0.139 |
| Levenshtein (all-to-all) | 0.440 | 0.391 | 0.373 | 0.005 | 0.203 | 0.176 | 0.026 | 0.004 |

### A.7 EFFECT OF INPUT SPECIFICATION FOR AUTOREGRESSIVE BASELINES

In the main experiments, multi-token prediction models (GRU-Dist, GRU-Matched, and GRU-Target) are trained in an all-to-one setting, where the model predicts the entire future window of target categories given the history with all features in datasets including timestamps. Autoregressive models, however, cannot be trained in an all-to-one formulation: their training objective requires predicting a single event at a time and conditioning each step on the previously generated outputs. Therefore, all autoregressive baselines are trained in the standard one-to-one setting. This architectural restriction leads to a mild asymmetry in the input setup.

To verify that this asymmetry does not influence our conclusions, we conducted an additional experiment in which the autoregressive baselines were trained in an all-to-all setting. In this variant, the model is provided with all future features in the prediction window, while the evaluation metrics (Matched-F1 and Levenshtein) are computed only on the target category, keeping the evaluation protocol unchanged.

Table 6 reports the comparison between the original one-to-one setup and the all-to-all variant. Across datasets, the all-to-all formulation does not improve autoregressive models. Crucially, the relative ranking between autoregressive and multi-token prediction approaches remains unchanged. This confirms that the advantages of GRU-Dist and related models are robust to the choice of input formulation.

### A.8 ADDITIONAL RESULTS

For completeness, we report all evaluation metrics across datasets. Levenshtein distance is included as an order-sensitive measure to quantify degradation under local shuffling (Figure 8), while the effect of shuffling on category diversity is illustrated by cardinality (Figure 9). The main text focuses on the order-invariant *Matched-F1 (micro)* (Figure 10), which we adopt as the primary evaluation metric throughout the study.

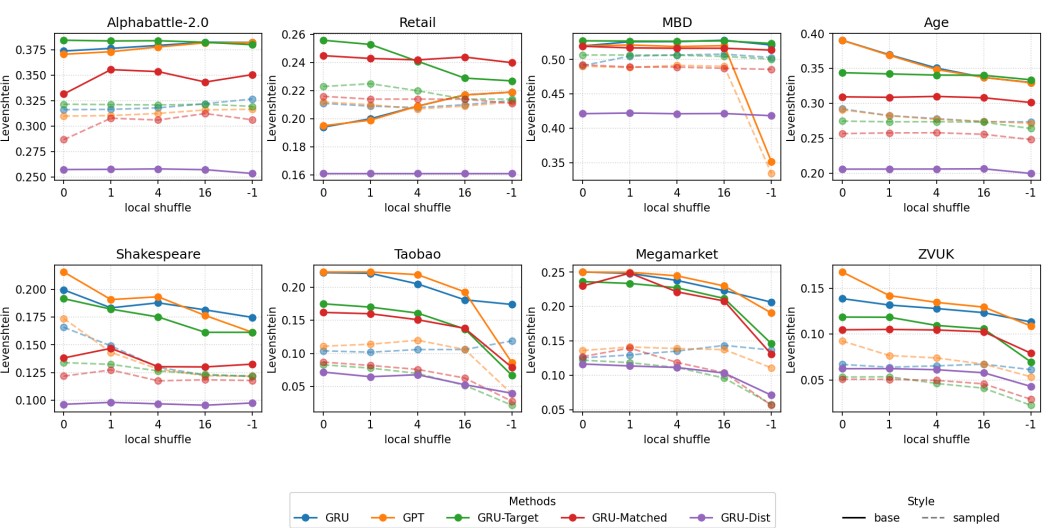

Figure 8: Levenshtein score on all datasets.

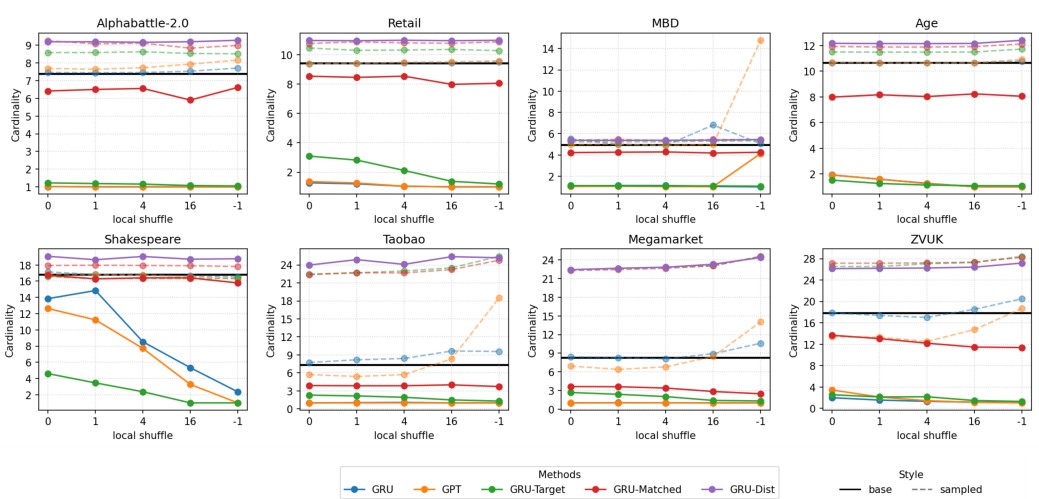

Figure 9: Effect of local shuffle on cardinality.

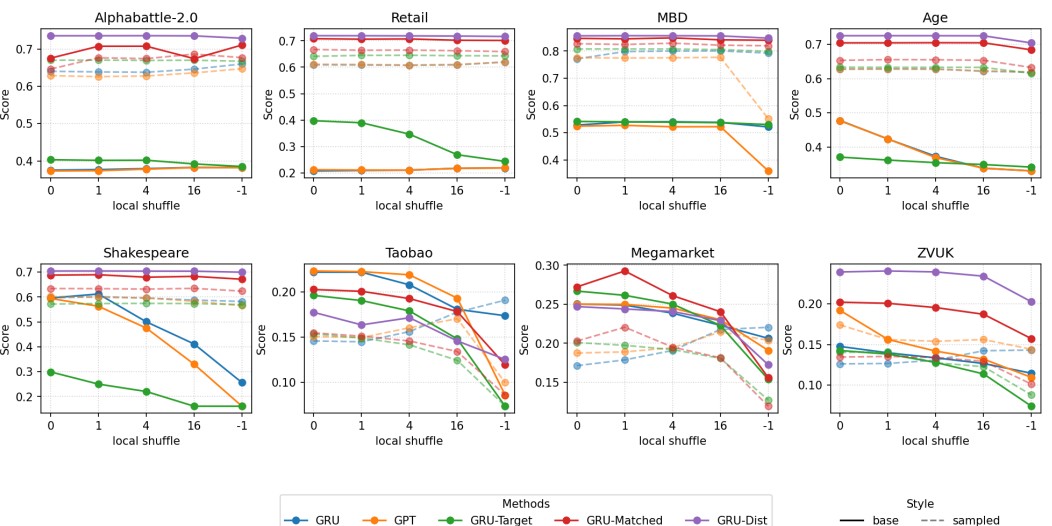

Figure 10: Next $N$ tokens forecasting. *Matched-F1 (micro)* results.

