# OpenReview forum: "HOW TO MODEL HUMAN ACTIONS DISTRIBUTION WITH EVENT SEQUENCE DATA"
_ICLR.cc/2026/Conference — ICLR 2026 Conference Desk Rejected Submission_

### Official Review · Reviewer_FL9s · 2025-10-27

**Soundness:** 3
**Presentation:** 2
**Contribution:** 3
**Rating:** 8
**Confidence:** 3

**Summary:**

This paper tackles a relevant question for sequence modelling tasks, especially in the era of LLMs and Transformers for non-NLP tasks. It investigates the central question of whether next-token prediction is the best paradigm for learning such models or whether the choice of learning task should be dataset / domain specific. In particular it introduces methods for identifying the order dependence of a dataset via the static-ness of the distribution of tokens in sliding windows over sequences.

It thoroughly evaluates how order dependency varies across domains and how various learning paradigms (next token prediction, multi token prediction, and direct future distribution prediction) perform in such domains.

**Strengths:**

Overall this is an interesting paper that answers a relevant question regarding whether next token prediction should be the learning task for large models across domains. The experiments are extensive, with 4 reasonable baselines, 3 learning paradigms, and 4 different approaches to learning future event distributions (autoregressive loss, target loss, matched loss, and order-invariant loss). The authors provide analysis of the static-ness of various datasets with quantitative results using a novel metric.

**Weaknesses:**

The authors could provide more exploration of and connection to previous work in quantifying distribution shift and static-ness of sequential data. Given the breadth of the experiments the authors undertake it would be helpful for them to tie them together - right now some things feel underexplained - for example what is the parameter lambda; how is section 4.3 connected to the rest of the paper; what is the GPT result reported alongside GRU everywhere?

**Questions:**

1) Why the choice of R=3 for the static-ness metric: is there analysis of the effect of R on the value of the static-ness metric, as this seems like a low number of anchor points.
2) The connection of 4.3 to the rest of the paper is not entirely clear - how are these windows constructed? Centered windows are constructed at each position and then within that window random shuffling is applied? It sounds like these windows overlap if this is performed at each position. Is Fig. 2 the perplexity from a fixed, pre-trained model on the shuffled sequences? Or is it the perplexity achieved by training a model on shuffled sequences?
3) What is λ - it is never explained and its relevance is not explored in the text?

---

> ### Author Response · Authors · 2025-11-21
> **Authors Rebuttal**
>
> **Thank you for your review and helpful comments. We note that the full codebase will be released with the camera-ready version.**
>
> ---
>
> ## W1: The authors could provide more exploration of and connection to previous work in quantifying distribution shift and static-ness of sequential data.
>
> That’s truth that some previous works have explored various forms of temporal distribution shift in sequential data, including proposed concept of context-driven distribution drift [1] seasonal shifts [2](Wu et al., SIGIR 2025 IDEA), as well as temporal dataset shift benchmarks [3] (Nguyen et al., NeurIPS 2022). These studies demonstrate that changes in the token distribution can significantly affect forecasting performance, which is closely aligned with our motivation. In the revised version, we have added references of these works in Section 4.2.
>
>
>
> [1] Chen, M., Shen, L., Fu, H., Li, Z., Sun, J., & Liu, C. (2024, August). Calibration of time-series forecasting: Detecting and adapting context-driven distribution shift. In Proceedings of the 30th ACM SIGKDD Conference on Knowledge Discovery and Data Mining (pp. 341-352).
>
>
> [2] Liao, Y., Yang, Y., Hou, M., Wu, L., Xu, H., & Liu, H. (2025, July). Mitigating distribution shifts in sequential recommendation: An invariance perspective. In Proceedings of the 48th International ACM SIGIR Conference on Research and Development in Information Retrieval (pp. 1603-1613).
>
>
> [3] Yao, H., Choi, C., Cao, B., Lee, Y., Koh, P. W. W., & Finn, C. (2022). Wild-time: A benchmark of in-the-wild distribution shift over time. Advances in Neural Information Processing Systems, 35, 10309-10324.
>
> ---
>
> ## W2: what is the GPT result reported alongside GRU everywhere?
>
>
> We include Transformer-based (GPT) results alongside GRU because these two architecture families remain the most competitive and widely used for event sequence forecasting. Prior work has shown that their relative performance depends on the task formulation (EBES [1], SeqNAS [2]) report that RNN-based models outperform Transformers on EvS-style classification task, while (HT-Transformers [3]) demonstrates that architectural modifications can close this gap for classification but do not translate into improved forecasting of future tokens. Since our work focuses specifically on the forecasting setting, we report results from both GRU- and GPT-based models to provide a representative comparison between the two dominant paradigms.
>
> At the same time, our goal is not to conduct a comprehensive study of “Transformers vs. RNN”. Such an investigation would require a dedicated analysis and is outside the score of this paper. The GPT results serve primarily as a strong Transformer-based reference, ensuring that our conclusions about order-invariant forecasting are not tied to a single model family.
>
> [1] Osin, D., Udovichenko, I., Shvetsov, E., Moskvoretskii, V., & Burnaev, E. (2025, August). Ebes: Easy benchmarking for event sequences. In Proceedings of the 31st ACM SIGKDD Conference on Knowledge Discovery and Data Mining V. 2 (pp. 5730-5741).
> [2] Udovichenko, I., Shvetsov, E., Divitsky, D., Osin, D., Trofimov, I., Sukharev, I., ... & Burnaev, E. (2024). SeqNAS: Neural architecture search for event sequence classification. IEEE Access, 12, 3898-3909.
> [3] Karpukhin, I., & Savchenko, A. (2025). Ht-transformer: Event sequences classification by accumulating prefix information with history tokens. arXiv preprint arXiv:2508.01474.
>
> ---
>
> ## Q1: Why the choice of R=3 for the static-ness metric: is there analysis of the effect of R on the value of the static-ness metric, as this seems like a low number of anchor points.
>
>
> We chose 𝑅 = 3 as a practical default based on a small sensitivity study conducted during the metric design. We observed that varying the number of anchor windows does not affect the dataset ranking or the relative scale of the staticity values.
>
> ---

---

> > ### Author Response · Authors · 2025-11-21
> >
> > ## Q2.1: How are these windows from 4.3 constructed?
> >
> > We have clarified this in the revised version, expanded Section 4.3 and added a reference in the results section to Section 4.3 to improve clarity.
> >
> >  “In Section 4.3, the local permutation procedure operates as follows. For each sequence position, we construct a symmetric window of radius 𝑤, and events are randomly permuted only within this window. These windows indeed overlap, which is intentional: the goal is to measure how robust the model is to progressively larger disruptions of local order. Formally, we implement the permutation via a restricted linear assignment: for each position 𝑖, only target positions j satisfying ∣i−j∣≤w are allowed, while all other assignments are masked out. Padding positions are also excluded, and the historical part and the target part of the sequence are shuffled independently to avoid leakage between them. This procedure produces a random but locally constrained permutation for every sequence in the batch.”
> >
> > ---
> >
> > ## Q2.2:  Is Fig. 2 the perplexity from a fixed, pre-trained model on the shuffled sequences? Or is it the perplexity achieved by training a model on shuffled sequences?
> >
> > We have also clarified it in Section 4.3:
> > “Fig. 2 shows the perplexity of models trained under the corresponding shuffling level. For each window size in Fig. 2, we train a separate model on sequences shuffled with that window and evaluate it on the original, unshuffled test data.“
> > Additionally, Fig. 2 illustrates that different datasets react very differently to the destruction of local temporal structure: banking domains remain largely unaffected, whereas datasets with strong local dependencies such as Shakespeare or Zvuk,  exhibit a sharp degradation as the window size increases.”
> >
> > ---
> >
> > ## Q3: What is λ - it is never explained and its relevance is not explored in the text?
> >
> > We have clarified the definition and role of the parameter λ in the revised version. Specifically, λ is the rate parameter of an exponential distribution fitted to the empirical histogram of event-category frequencies. It reflects how quickly category frequencies decay and therefore serves as a quantitative indicator of distributional imbalance in each dataset. A detailed description is now included in Section 4.2. We have also appended references to Figure 1 and Table 2.
> >
> > ---
> >
> > ## Overall
> >
> > **We sincerely thank the reviewer for their thoughtful comments and hope we have addressed the concerns raised. If the reviewer is satisfied with our responses and finds the work has strong merit for publication, we would greatly appreciate a score reconsideration.**

---

### Official Review · Reviewer_aqT3 · 2025-10-31

**Soundness:** 2
**Presentation:** 3
**Contribution:** 2
**Rating:** 4
**Confidence:** 3

**Summary:**

This paper investigates the performance of different forecasting models applied to event sequence data. Specifically, it challenges the prevailing autoregressive paradigm and examines models that explicitly represent the future distribution of events, independent of their temporal order. The central assumption underlying this work is that, in certain applications, the precise chronological ordering of events may not be essential for accurate prediction. To address this perspective, the authors propose a new formulation that focuses on predicting the distribution of future events within a specified temporal window. The study empirically compares four distinct prediction formulations: (1) predicting the next single event, (2) predicting the sequence of subsequent events, (3) predicting the set of subsequent events, and (4) predicting the distribution of subsequent events. The experimental results suggest that, on average, order-invariant models outperform order-based models. However, it is important to note that the evaluation relies on an order-invariant metric, which may inherently favor such models. The paper further introduces the staticity index, a novel measure designed to estimate the degree to which event order matters within a dataset. This index can assist in determining whether an order-based or order-invariant modeling paradigm is more suitable for a given application.

**Strengths:**

- The proposed concept of predicting the distribution of actions is interesting and potentially impactful. This formulation appears to be novel and is particularly relevant in scenarios where the local temporal order of events is not essential for accurate forecasting. The model is trained by optimizing the KL divergence between the predicted distribution and the empirical event distribution, which provides a principled approach for aligning the model’s outputs with the observed data.

- The paper introduces a staticity index designed to estimate the degree to which the sequence order is important within a dataset. The staticity index quantifies how the empirical distribution of events within each sequence evolves over time. By capturing this temporal stability or variability, the index can serve as a diagnostic tool for determining whether an order-based or order-invariant problem formulation is more appropriate for a given dataset.

- The empirical evaluation includes a comparison of four predictive formulations across eight benchmark datasets. The four problem formulations are: (1) predicting the next single event, (2) predicting the sequence of subsequent events, (3) predicting the set of subsequent events, and (4) predicting the distribution of subsequent events. The proposed approach demonstrates superior performance on five out of the eight datasets. These results suggest that when the temporal ordering of future events carries limited predictive information, modeling the distribution of event types alone can be sufficient for achieving strong forecasting performance.

- The paper also introduces an order-invariant Matched-F1 score, an evaluation metric that treats sequences as unordered sets (bags) of events. This metric enables fair comparison between order-sensitive and order-invariant models, particularly in cases where the temporal order may not be relevant.

- The paper is easy to read and understand. It provides enough context to understand the problem and the key contributions.

**Weaknesses:**

- The paper would benefit from a more comprehensive analysis of the proposed GRU-dist model. Since this model represents the main model contribution, a deeper examination of its behavior and performance is essential to strengthen the paper’s empirical findings. For instance, it would be informative to provide a detailed comparison between GRU-dist and GRU-matched across several datasets, highlighting where and why the performance differences occur. Additionally, the paper could expand on the analysis of the sampling process used in order-invariant models.

- The paper states that it introduces “a KL-based metric to quantify temporal drift,” but the conceptual link between the proposed metric and the KL divergence is not clearly explained or sufficiently justified. Providing a clearer theoretical motivation and detailing how the metric relates to the KL formulation would help the reader better understand its validity and relevance. Including an extended explanation or derivation in the supplementary material would be a suitable way to address this.

- Because of the specific problem formulation adopted in the paper, the model does not evaluate the temporal prediction component typically considered in the standard TPP framework. The time-invariant formulation focuses exclusively on predicting event distributions rather than event timings. This distinction limits the direct comparability of the proposed method with traditional time-sensitive TPP models.

- The evaluation metric introduced in the paper appears to favor order-invariant models, since it does not account for the temporal ordering of events. This bias could lead to an unfair comparison between order-based and order-invariant approaches. Models that are designed to capture event order are inherently penalized when evaluated using an order-agnostic metric. It would therefore be valuable for the paper to discuss this limitation and, if possible, include additional metrics that can more equitably assess both model types.

**Questions:**

- As mentioned earlier, the paper would benefit from a deeper analysis of the performance and behavior of the proposed approach. Providing a more thorough interpretation of the results would help clarify the strengths and limitations of the model, as well as the conditions under which it performs best.

- In Table 2, the meaning of the symbol S is not explained. The table caption should explicitly define this notation to ensure that the results are self-contained and easy to interpret without referring back to the main text.

- In Table 3, it is unclear how many next events are being predicted. The caption currently mentions only N, but not its value.

- In Table 3, it would be valuable to include an explanation or discussion of why the Repeat method performs better than other methods on three of the datasets. Understanding the reasons behind this result could provide useful insights into the characteristics of the datasets or the modeling assumptions that favor this method.

- Also in Table 3, it would be helpful to explain why the GT performance for the Taobao dataset is not equal to 1.0.

---

> ### Author Response · Authors · 2025-11-21
> **Authors Rebuttal**
>
> **We thank the reviewer for their careful reading and constructive feedback. We also note that the full codebase will be released with the camera-ready version.**
>
> Addressing weaknesses:
>
> ## W1. Insufficient analysis of GRU-dist
> We substantially expanded Section 7 (Results) with a direct comparison of GRU-Dist vs. GRU-Matched, including dataset-specific explanations for performance gaps (e.g., high CRR in Taobao/Megamarket limits GRU-Dist’s ability to exploit local repetition). We have also added a detailed description of the sampling procedure for GRU-Dist (Section 6.3)
> All changes are reflected in the revised manuscript.
>
> ---
> ## W2. Misleading description of the metric as KL-based without proper justification.
> We acknowledge this oversight; the metric is distribution-based, not KL-based. We have corrected the terminology in the introduction and expanded its description in Section 4.2 for clarity.
>
> ---
> ## W3. The evaluation excludes timing prediction, limiting fair comparison with time-sensitive TPP models.
> Our focus is deliberately on distribution forecasting, a distinct but practically relevant task where timing is often irrelevant; comparing against TPP SOTAs would conflate event-type and time modeling, and extending GRU-Dist to joint category–time prediction remains an important but separate challenge. Extending to full TPP is future work.
>
> ---
> ## W4. Unfair comparison between order-based and order-invariant approaches
> We acknowledge that Matched-F1 favors order-agnostic models by design, as our task is distribution forecasting. To ensure balanced evaluation, we now report and analyze Levenshtein distance (Section 7, “Levenshtein performance”) and explicitly discuss the metric trade-off. Our analysis shows that on many datasets, NTP methods perform only marginally better than Mode baseline, indicating little benefit from modeling order. Only under strong local structure (high CRR, low Staticity) do NTP models meaningfully outperform.
>
> ---
> ## Q1. In Table 2, the meaning of the symbol S is not explained
> The caption of Table 2 has been updated to include a brief definition of the Staticity index: “$S$: average distributional similarity over time (1 = fully stationary).”
>
> ---
> ## Q2. In Table 3, it is unclear how many next events are being predicted.
> We have explicitly added N=32 to the caption of Table 3 (and all relevant places) for clarity.
>
> ---
> ## Q3. Why the Repeat method performs better than other methods on three of the datasets
> We have added this explanation to the Results section: Repeat excels on Taobao, Megamarket, and Zvuk because these datasets exhibit extremely low λ (near-uniform category frequencies) and very high cardinality, making distribution estimation unstable.
>
> ---
> ## Q4. Why the GT performance for the Taobao dataset is not equal to 1.0.
> The GT Matched-F1 on Taobao is < 1.0 due to the category clipping preprocessing adopted from EBES (Osin et al., 2025), where rare categories are merged into a single token, introducing a small, fixed information loss even for ground truth. This affects all methods equally and does not impact relative comparisons or conclusions.
>
> ---
> ## Overall
> We sincerely thank the reviewer for their thoughtful comments and hope we have addressed the concerns raised. If the reviewer is satisfied with our responses and finds the work has strong merit for publication, we would greatly appreciate a score reconsideration.

---

> > ### Comment · Reviewer_aqT3 · 2025-11-26
> >
> > Thank you for the clarifications provided in your responses. They address most of my earlier concerns, and I will adjust my score accordingly. However, I remain unconvinced that the comparison between order-based and order-invariant approaches is fully fair. I understand that predicting temporal information is not central to the scope of this paper, and I am willing to set that point aside. My main concern is that the order-based approaches appear to be evaluated in a way that does not align with the objectives for which they are trained. These models are designed to capture more complex structural patterns by predicting the order of actions, which is an inherently more demanding task. Despite this, the evaluation does not measure how well they perform on this part of the problem. Consequently, the order-based methods are optimized for a different objective than the one used for the final evaluation metrics. This mismatch may lead to an underestimation of their capabilities and could make the comparison inherently unfavorable to them. To increase my score further, I would need to see a more balanced comparison protocol.

---

> > > ### Author Response · Authors · 2025-12-01
> > > **Author Response**
> > >
> > > We appreciate the reviewer’s thoughtful concern and clarify our intent:
> > >
> > > 1. We do **not** claim that order-based methods (e.g., NTP) are universally inferior. Rather, for the *task of forecasting future event distributions*, order-invariant objectives often suffice, and *outperform* NTP when evaluated on *task-aligned metrics* (e.g., Matched-F1). NTP was included as the de facto baseline to caution against its uncritical use in domains where order is weakly informative.
> > >
> > > 2. As shown in Tables 3–4 and Figure 3, on datasets with weak temporal structure (MBD, Age, AB, Retail), NTP models suffer severe mode collapse (Cardinality < 2) and perform nearly identically to the *Mode* baseline, even under Levenshtein. This suggests that **in such settings, order-sensitive metrics may not meaningfully differentiate models**, as the “order” being measured is effectively noise. This is why Matched-F1 is our primary metric -- it aligns with the task goal.
> > >
> > > 3. We fully report and discuss Levenshtein (Section 7), explicitly stating: *“As expected, order-agnostic models perform worse, since they impose no ordering constraints.”* We do not obscure this trade-off; instead, we frame model choice as *task- and data-dependent*.
> > >
> > > We’re happy to add a clarifying sentence in the Conclusion (e.g., *“Our results do not dispute the value of order-based modeling in inherently sequential domains (e.g., language), but emphasize that its superiority is not guaranteed, and must be validated per use case.”*) if the reviewer finds it helpful.

---

### Official Review · Reviewer_6xsi · 2025-11-10

**Soundness:** 2
**Presentation:** 2
**Contribution:** 1
**Rating:** 2
**Confidence:** 2

**Summary:**

Forecasting human action sequence is an important task in various domains such as retail, finance, and recommendation.
An autoregressive model is a dominant solution in the field due to its huge success in NLP,
yet they have limitations in this domain to address distributional imbalance or order irrelevance.
This paper studies how does GRU handle human action sequence forecasting.

**Strengths:**

*	Local order irrelevant evaluation seems important in real-world situations.
*	Authors conduct experiments in various datasets

**Weaknesses:**

*	The three hypotheses and four-step framework presented in the introduction are not sufficiently discussed or supported in the later sections.
*	The novelty of the work appears limited. The main idea—modifying the GRU loss function to hit one of the k consecutive targets instead of a single one—seems rather trivial, given the stated goal of evaluating continuous k-hit performance.
*	The paper lacks implementation details or released code to ensure reproducibility.

**Questions:**

*	What is the core contribution of this paper? Does it propose a new problem setting or a new method, or is it mainly an empirical study applying existing techniques to a new context?
*	Are the actual timestamps in the sequential data considered in the model or evaluation?
*	While experimental results are provided for GRU and its variants, what about Transformer-based models?

---

> ### Author Response · Authors · 2025-11-21
> **Authors Rebuttal**
>
> **Thank you for your review. We also note that the full codebase will be released with the camera-ready version.**
>
> ---
>
> ## W1. The three hypotheses are not sufficiently supported in the later sections.
> We agree that hypotheses were not clearly discussed in later sections. To add clarity to the paper we revised the Introduction and Conclusion section. However, framing a hypothesis that turns out to be  inconsistent with the data  is essential to the research process [1,2,3,4]. When we reject hypotheses we actually gain knowledge. So, we do not consider it as a weakness.
>
>
>
> [1] Platt, J. R. (1964). Strong inference. Science, 146(3642), 347–353. https://www.science.org/doi/pdf/10.1126/science.146.3642.347  <br>
>
> [2] Chamberlin, T. C. (1890/1965). The method of multiple working hypotheses. Science, 148(3667), 754–759. https://www.science.org/doi/pdf/10.1126/science.148.3667.754  <br>
>
> [3] Ioannidis, J. P. A. (2005). Why most published research findings are false. PLOS Medicine, 2(8), e124. https://journals.plos.org/plosmedicine/article/file?id=10.1371/journal.pmed.0020124&type=printable  <br>
>
> [4] Nosek, B. A., Ebersole, C. R., DeHaven, A. C., & Mellor, D. T. (2018). The preregistration revolution. Proceedings of the National Academy of Sciences, 115(11), 2600–2606. https://www.pnas.org/doi/pdf/10.1073/pnas.1708274114
>
> ---
>
> ## W2. Novelty and core contribution.
>
> We would like to clarify that the core contribution of our work is not a modification of the GRU loss. The primary novelty lies in (1) systematically studying explicit distribution forecasting for event sequences as a distinct task, (2) demonstrating when such explicit modeling should replace autoregressive objectives, and (3) establishing dataset-level diagnostics (Staticity Index, Local Permutation Analysis, λ, CRR) that predict when temporal order is informative. The GRU-Dist objective is only one of the contributions.
>
> To make this clear, we have substantially revised the Introduction to emphasize the problem formulation, and the conditions under which explicit distribution modeling is preferable to NTP/MTP. The revised text now precisely states the contribution beyond a model-level modification.
>
> ---
>
> ## W3. The paper lacks implementation details or released code to ensure reproducibility.
>
>
> We will release code upon publication.
>
> ---
>
> ## Q1. Are the actual timestamps in the sequential data considered in the model or evaluation?
>
>
> Multi-token models (GRU-Dist, GRU-Matched, GRU-Target) are trained in an all-to-one setting, predicting the full future window from the complete history (including timestamps). Autoregressive models cannot use all-to-one, since they must generate events token-by-token, conditioning each step on their own previous outputs; therefore, they are trained in the standard one-to-one mode.
> To confirm that this asymmetry does not influence the results, we additionally trained all autoregressive baselines in an all-to-all setup. Even with access to the full history and all future features, their performance does not improve (and often degrades), and the relative ranking of methods remains unchanged.
>
> We added these results and details in Appendix A.7.
>
> ---
>
> ## Q2. While experimental results are provided for GRU and its variants, what about Transformer-based models?
> Yes, as one of our architectures we use GPT architecture, which is transformer based architecture, we refer to it as GPT in our experiments.
>
> ---
>
> **We sincerely thank the reviewer for their thoughtful comments and hope we have addressed the concerns raised. If the reviewer is satisfied with our responses and finds the work has strong merit for publication, we would greatly appreciate a score reconsideration.**

---

### Author Response · Authors · 2025-11-21
**Updated manuscript.**

Dear Reviewers and Area Chair,

Thank you for your thoughtful feedback on our manuscript. In response to your comments, we have carefully revised the paper to address all concerns raised.  **The revised manuscript has been uploaded to the submission system with tracked changes in yellow color**.

---

### Note · Program_Chairs · 2026-01-17
**Submission Desk Rejected by Program Chairs**

The following references in this submission do not refer to real documents and/or have major errors in bibliographic information:

 Steffen Rendle. Recommender systems. Foundations and Trends in Information Retrieval, 14(1-2)